# The Effect of an Alternating Magnetic Field-Assisted Freezing Process on the Quality of Frozen Penaeus Japonicus

**DOI:** 10.3390/foods14234112

**Published:** 2025-11-30

**Authors:** Dazhang Yang, Qifei Zhang, Jing Xie, Guoqing Ding

**Affiliations:** 1College of Food Science and Technology, Shanghai Ocean University, Shanghai 201306, China; dzyang@shou.edu.cn (D.Y.); qifeizhang11@outlook.com (Q.Z.); m220300860@st.shou.edu.cn (G.D.); 2Shanghai Professional Technology Service Platform on Cold Chain Equipment Performance and Energy Saving Testing Evaluation, Shanghai 201306, China; 3National Experimental Teaching Demonstration Center for Food Science and Engineering, Shanghai Ocean University, Shanghai 201306, China; 4Quality Supervision, Inspection and Testing Center for Cold Storage and Refrigeration Equipment, Ministry of Agriculture, Shanghai 201403, China

**Keywords:** ice crystal size, microstructure, water-holding capacity, alternating magnetic field-assisted freezing

## Abstract

Freezing is a common food preservation method, but conventional freezing often produces large, irregular ice crystals that damage muscle tissue and degrade food quality. This study developed an experimental system using an impact freezer to investigate the effects of alternating magnetic fields (AMFs) of different intensities (0 G, 20 G, 40 G, 60 G, and 80 G) and frequencies (50 Hz, 100 Hz, 150 Hz, 200 Hz, and 250 Hz) on the freezing behavior and muscle quality of Penaeus Japonicus. Results showed that applying a 40 G AMF (AMF-40) significantly reduced freezing time, thawing loss, and cooking loss. It also improved water retention, texture, and color stability. Water distribution analysis indicated that AMF-40 limited the movement and loss of immobilized and free water. Microstructural observations revealed smaller pores and more intact muscle fibers, suggesting the formation of finer ice crystals. Under a 200 Hz AMF (AMF-200 Hz), samples exhibited further decreases in freezing time, thawing loss, and cooking loss, along with significant improvements in hardness, and Springiness, while maintaining muscle color. Enhanced water-holding capacity was also observed, preserving bound water content. Overall, both AMF-40 and AMF-200 Hz promoted the formation of smaller ice crystals and effectively preserved the muscle quality of Penaeus Japonicus during freezing, improving the preservation outcome.

## 1. Introduction

Penaeus Japonicus is notable for its high content of astaxanthin, amino acids, eicosapentaenoic acid (EPA), and essential trace elements such as zinc and iron. The primary approach for preserving the quality of shrimp and other aquatic products involves the application of low-temperature technologies. Exposure to low temperatures inhibits microbial metabolism and significantly suppresses enzymatic activity in food products [1]. However, ice crystal formation during the freezing process can detrimentally affect the quality of frozen products [2], as these crystals may damage muscle fibers and reduce the nutritional value of the samples. To mitigate these adverse effects, researchers are investigating the integration of physical fields—such as ultrasound, high pressure, and microwave energy—with conventional freezing methods [3]. Magnetic field-assisted freezing is one such approach, involving the application of magnetic fields with varying intensities or frequencies in conjunction with traditional freezing techniques [4]. This method has demonstrated potential to accelerate the freezing process, reduce ice crystal size, improve the quality of frozen aquatic products, and extend their shelf life [5].

The mechanism of the influence of magnetic fields on food is still incomplete, as indicated by previous studies. Some researchers have proposed that exposure to magnetic fields can alter the growth, viability, and gene expression of specific bacteria, potentially leading to a reduction in microbial populations and slowing the spoilage processes of food [6]. When temperatures fall below a critical threshold or when thermal vibrations are suddenly diminished, instantaneous crystallization may occur, resulting in the formation of numerous small ice crystals. Moreover, as illustrated in Figure 1, the application of a magnetic field strengthens the hydrogen bonds, causing the weaker bonds to break, which results in larger clusters of water molecules breaking into more tightly bound, smaller clusters. Consequently, during the freezing phase transition, this leads to the formation of multiple small ice crystals, thereby minimizing damage to the cellular structure of the sample [7].

Research on magnetic field-assisted freezing technology has predominantly focused on fruits [8,9,10], vegetables [3,11], starch-based products [12], and livestock meat [13,14,15,16]. Different frozen items exhibit distinct optimal magnetic field intensities. Similarly, Purnell and Kim conducted a similar study, which yielded comparable results [9,13]. While numerous studies have investigated the effects of magnetic fields on food preservation, relatively few have concentrated on aquatic products, such as shrimp. Due to differences in moisture content, carbohydrate composition, and protein structure across various foods, their responses to magnetic fields vary accordingly. As a result, the optimal magnetic field strengths for frozen foods are likely to differ depending on the specific characteristics of the product [13].

To investigate the effects of varying magnetic field intensities and frequencies on the muscle characteristics of frozen Penaeus Japonicus, an experimental protocol was developed encompassing a low Magnetic Field Strength range (0–80 G) and a frequency range of 50–250 Hz. As illustrated in Figure 2, this study examined the effects of varying magnetic field intensities and frequencies on temperature dynamics, microstructural integrity, moisture distribution, and physicochemical properties of Penaeus Japonicus during the freezing process. The objective was to identify the optimal magnetic field parameters capable of enhancing the quality of frozen Penaeus Japonicus.

## 2. Materials and Methods

### 2.1. Magnetic Field-Assisted Freezing System

The magnetic field-assisted freezing system consists of a magnetic field generator, a freezing unit, and a monitoring system, as illustrated in Figure 3. The magnetic field generator comprises Helmholtz coils and an AC power supply. The Helmholtz coils generate a magnetic field through the flow of electric current, in accordance with Ampère’s law. When current passes through the coils, each coil produces a magnetic field, and their overlapping fields create a relatively uniform magnetic field in the central region. The Penaeus Japonicus samples were positioned in this central area to ensure exposure to the uniform alternating magnetic field generated by the Helmholtz coils. An alternating voltage was applied to the coils to induce the alternating magnetic field. The model of the Helmholtz coil and the magnetic field strength at the freezing position of Penaeus Japonicus are shown in Figure 4a. The AC power supply used in the experiment was an IVYTECH APS4000C (IVYTECH, Shenzhen, China), with a rated power of 1200 W, an adjustable voltage range of 0–150 V, and a current range of 0–10 A. In the experiment conducted to ascertain the magnetic field strength, the alternating magnetic field’s frequency was set to 50 Hz. In the magnetic field frequency experiment, the intensity of the alternating magnetic field was set to 40 G. The uniformity of the magnetic field was simulated using COMSOL Multiphysics 6.3, as shown in Figure 4b. Although the uniformity of the magnetic field may be slightly influenced by factors such as coil winding techniques, it remains generally stable. Furthermore, the electric field distribution was simulated using COMSOL, as depicted in Figure 4c. The results indicate that the generated electric field strength is negligible.

The freezing unit consisted of a cold storage chamber and an impact freezer. The impact freezer directed airflow onto the samples at a wind speed of 2 m/s. The monitoring system included temperature monitoring and magnetic field strength detection. A T-type thermocouple and a data acquisition device were used to measure the temperature at the sample center, with real-time data displayed on a computer. The uniformity and strength of the magnetic field were verified using a TM5100A Digital Tesla Meter (TUNKIA, Changsha, China). The magnetic field strength was monitored indirectly through current measurements.

### 2.2. Pre-Treatment of Shrimp

Uniformly sized live samples of Penaeus Japonicus (25–35 individuals per 500 g) were purchased from the Luchao Harbor Aquatic Market in Shanghai, China. They were transported to the laboratory in oxygenated water. The shrimp were promptly euthanized and decapitated, and representative samples with an average weight of 10 ± 0.5 g and a body length of 10 ± 0.5 cm were selected. The untreated fresh samples served as the control group, while the remaining samples were divided into ten experimental groups for freezing under different magnetic field strengths and frequencies, with each group containing three Penaeus japonicus. These shrimp were selected from the same harvest batch and matched for body size, growth conditions, and physiological status. This controlled sample composition was intended to minimize inter-individual biological variation that could otherwise confound the effects of magnetic field parameters on the freezing process. Although the number of shrimp per sample was limited, the use of multiple independent replicates ensured the robustness and statistical reliability of the results. Magnetic field strengths were set at 0 G (AMF-0), 20 G (AMF-20), 40 G (AMF-40), 60 G (AMF-60), and 80 G (AMF-80). Magnetic field frequencies were set at 50 Hz (AMF-50 Hz), 100 Hz (AMF-100 Hz), 150 Hz (AMF-150 Hz), 200 Hz (AMF-200 Hz), and 250 Hz (AMF-250 Hz). After adjusting the voltage, samples were frozen once the current and magnetic field stabilized. When the core temperature of the samples reached −18 °C, they were transferred to a −18 °C freezer and stored for three days. Five groups of alternating magnetic fields with different intensities and five groups of alternating magnetic fields with different frequencies were established to investigate their effects on the freezing quality of Penaeus Japonicus. The groups having magnetic field strengths of 0 G, 20 G, 40 G, 60 G, and 80 G are designated as AMF-0, AMF-20, AMF-40, AMF-60, and AMF-80. The groups having magnetic field frequencies of 50 Hz, 100 Hz, 150 Hz, 200 Hz and 250 Hz are designated as AMF-50 Hz, AMF-100 Hz, AMF-150 Hz, AMF-200 Hz and AMF-250 Hz, with ‘Fresh’ indicating the fresh group.

### 2.3. Determination of Freezing Curves

During the freezing process, a T-type thermocouple probe was inserted into the geometric center of each shrimp sample to monitor the internal temperature. The probe was connected to a data acquisition device, which was subsequently linked to a computer. Dedicated software was used to monitor and record the central temperature of the samples in real time, with a data logging frequency of one measurement per second.

### 2.4. Scanning Electron Microscopy

To examine microstructural changes, muscle fibers were extracted from the abdominal region of the shrimp samples for observation under an electron microscope. To prevent thawing, the samples were sectioned into dimensions of 5 mm× 5 mm × 2 mm within a −25 °C freezer. Subsequently, they were transferred to a −80 °C freezer using a low-temperature storage container. The samples were pre-cooled at −80 °C for 24 h, followed by being freeze-dried in a freeze-dryer (FDU-2110, EYELA, Shanghai, China) for 48 h. A thin layer of gold was then applied to the samples using a gold–palladium sputter coater. The microstructure of Penaeus Japonicus was observed using a scanning electron microscope (SU5000, Hitachi, Tokyo, Japan) operated at an accelerating voltage of 5 kV and a magnification of 300×.

### 2.5. Determination of Thawing Loss, Cooking Loss, and Centrifugal Loss

The samples were thawed in a conventional refrigerator at 4 °C until the core temperature reached 4 °C. The shrimp were weighed before and after thawing (denoted as W_0_ and W_1_, respectively), with surface moisture gently removed using filter paper. Each measurement was conducted in triplicate, and the average value was used for analysis. Thawing loss was calculated using the following equation:(1)$$\mathrm{Thawing}\;\mathrm{loss}\;(\%)\;=\frac{W_{0}-W_{1}}{W_{0}}\times 100\%$$

Thawed shrimp samples were weighed to approximately 2.0 ± 0.5 g, wrapped in filter paper, sealed in polyethylene bags, and steamed in a water bath at 80 °C for 6 min. After steaming, the samples were gently dried using filter paper and weighed both before and after the steaming process (denoted as W_2_ and W_3_, respectively). The weights of each sample were recorded for further analysis.(2)$$\mathrm{Cooking}\;\mathrm{loss}\;(\%)=\frac{W_{2}-W_{3}}{W_{2}}\times 100\%$$

To determine centrifugal loss, approximately 2.0 ± 0.5 g of thawed shrimp meat was weighed. Surface moisture was removed using filter paper, and the samples were then wrapped in two layers of filter paper. The samples were centrifuged at 5000 r/min at 4 °C for 10 min. The sample weights before and after centrifugation (denoted as W_4_ and W_5_, respectively).(3)$$\mathrm{Centrifugal}\;\mathrm{loss}\;(\%)=\frac{W_{4}-W_{5}}{W_{4}}\times 100\%$$

### 2.6. Low-Field Nuclear Magnetic Resonance (LF-NMR) and Magnetic Resonance Imaging (MRI) Analysis

An LF-NMR instrument (MesoMR23-060H-1, Niumai, Shanghai, China) was utilized to measure the relaxation time and conduct magnetic resonance imaging (MRI) of Penaeus Japonicus. Three samples, each weighing 2.0 ± 0.5 g and obtained from the first and second abdominal segments of three individual shrimp, were weighed, wrapped in cling film, and placed into a 60 mm diameter MRI detection tube. The transverse relaxation time (T_2_) curve was obtained through iterative inversion using analysis software, based on the exponential decay curve derived from the Carr–Purcell–Meiboom–Gill (CPMG) sequence. CPMG is a nuclear magnetic resonance (NMR) pulse sequence used for measuring the transverse relaxation time of a sample. The moisture distribution of the samples was visualized using Newmark imaging software 4.0, and the corresponding proton density maps were analyzed using Newmark image processing software 4.0.

### 2.7. Determination of Color

The color characteristics of the thawed shrimp samples were quantified using a colorimeter (YS6010) (3nh, Guangdong, China). After calibration with standard black and white plates, the instrument was placed vertically on the surface of the sample for measurement. Color values were recorded in the CIELAB color space, as defined by the International Commission on Illumination (CIE), using L*, a*, and b* parameters. Here, L* represents lightness, a* indicates the red–green axis, and b* corresponds to the yellow–blue axis. In the evaluation of freezing quality for Penaeus Japonicus, the L* value is primarily used to assess color changes. One measurement was taken from each frozen shrimp sample, and three individual samples from each group were tested once. Meanwhile, to comprehensively quantify these observed color changes, the total color difference (ΔE*) was calculated using the widely recognized CIE 1976 LAB color difference formula.

The formula is as follows:(4)$$\Delta E\;\times \;\mathrm{ab}\;=\;\sqrt{({\Delta L}^{*})²+({\Delta a}^{*})²+({\Delta b}^{*})²}$$
where ΔL* represents the brightness difference, indicating the variation in brightness between two colors. Δa* denotes the difference along the red–green axis, reflecting the chromatic variation between two colors on this axis. Δb* signifies the difference along the yellow–blue axis, representing the chromatic variation between two colors on this axis.

### 2.8. Measurement of pH

Weigh 2 g of the sample and add 18 mL of deionized water at 4 °C. The mixture is then homogenized for 2 min using a homogenizer. The sample is further ground and homogenized using a high-speed blender. The pH of the sample is measured using a pH meter.

### 2.9. Determination of Texture Properties

The texture of the sample was evaluated using a texture analyzer (SMS TA.XT Plus, SMS, Godalming, UK), specifically measuring hardness and springiness. The first and second segments of the shrimp bodies were weighed as 2 ± 0.5 g samples to evaluate textural changes. The testing conditions were as follows: pre-test rate of 2 mm/s, test rate of 1 mm/s, post-test rate of 2 mm/s, compression set at 50%, dwell time of 5 s, and ambient temperature ranging from 11 °C to 16 °C.

### 2.10. Determination of Differential Scanning Calorimetry (DSC)

The sample weight was measured using an electronic balance with a precision of 0.1 mg. A muscle sample weighing between 5 and 20 mg was then compressed in a crucible and analyzed using a differential scanning calorimeter (TA Instruments, New Castle, DE, USA). An empty aluminum crucible was sealed for comparison purposes. The samples were heated from 20 °C to 100 °C at a rate of 5 °C per minute. The maximum denaturation temperature (T_max) and the enthalpy of denaturation (ΔH) were determined from the thermograms using NanoAnalyze 3.11.

### 2.11. Statistical Analysis

The data were statistically analyzed using SPSS software (version 26.0, SPSS Inc., Chicago, IL, USA) and are presented as mean ± standard deviation. The significance of the main effects was evaluated using one-way analysis of variance (ANOVA) followed by Waller–Duncan post hoc testing at a significance level of *p* < 0.05. Data on the moisture characteristics and physicochemical properties of Penaeus Japonicus were analyzed using a mixed-model procedure, with triplicates treated as random effects and the freezing treatments (Fresh, AMF-20, AMF-40, AMF-60, AMF-80, AMF-50 Hz, AMF-100 Hz, AMF-150 Hz, AMF-200 Hz, and AMF-250 Hz) considered as fixed effects. All experiments were conducted in triplicate for each batch of Penaeus Japonicus.

## 3. Results and Discussion

### 3.1. The Effect of Magnetic Field Strength

#### 3.1.1. Changes in Freezing Time at Different Magnetic Field Strengths

Considering the potential influence of the magnetic field on thermocouple measurements, Figure 5a presents three repeated measurements of temperature variation recorded before and after the magnetic field was activated. The coil remained unenergized for the first 10 min. At the 10 min mark, the Helmholtz coil was energized to determine whether the activation induced any significant change in the temperature readings obtained from the thermocouple.

The mean and standard deviation of temperature fluctuation changes in the first test were calculated as −0.0004 ± 0.0338 °C. The thermocouple temperature change before and after the coil was powered on was 0.0092 °C, which remained within the normal fluctuation range. For the second test, the mean and standard deviation of the temperature fluctuation changes were calculated as −0.0002 ± 0.0339 °C. The thermocouple temperature change before and after powering on the coil was −0.0469 °C, also within the normal fluctuation range. The mean and standard deviation of the temperature fluctuation changes for the third test were calculated as −0.0006 ± 0.0353 °C. Before and after the coil was powered on, the thermocouple temperature change was 0.0072 °C, remaining within the normal fluctuation range. The results indicate that the application of a magnetic field does not significantly influence thermocouple readings, and the recorded temperature profiles remain within an acceptable range of fluctuation. In their study, Gou et al. [17] utilized a thermocouple as a temperature measurement instrument to investigate the effects of permanent magnets on droplet freezing and frost formation on cold surfaces.

The typical freezing curves observed in this experiment are presented in Figure 5b. Notably, no significant subcooling was detected in any of the experimental groups. According to the findings of Leng et al. [18], a pronounced subcooling phenomenon was observed during the freezing process at −18 °C. However, at −40 °C, subcooling was not evident due to the more rapid freezing process. This observation suggests that the cooling rate during freezing plays a crucial role in determining the stability of water in its liquid state. Under rapid freezing conditions, the temperature of the water decreases swiftly, preventing the formation of the conditions necessary for subcooling.

The freezing curve exhibits a temperature range from −18 °C to 4 °C. Based on the conditions of the freezing process, the curve can be divided into three distinct phases: the pre-cooling stage (4 °C to −1 °C), the phase transition stage (−1 °C to −5 °C), and the subcooling stage (−5 °C to −18 °C) [19]. Table 1 summarizes the duration spent by the samples at each stage based on typical freezing curves. Owing to the substantial temperature difference between the samples and the interior of the freezer, a rapid decline in temperature was observed during the pre-cooling phase. Specifically, the AMF-40 group exhibited a pre-cooling time that was 37.4% shorter than that of the AMF-0 group (*p* < 0.05). Among all experimental groups, the AMF-20 group exhibited the longest pre-cooling duration, which was 26 s longer than that of the AMF-0 group. These results indicate that the application of a magnetic field effectively reduced the pre-cooling time of Penaeus Japonicus. This effect may be attributed to the diamagnetic nature of water molecules. The reorientation of water molecules under a magnetic field is believed to enhance thermal conductivity, thereby accelerating the pre-cooling process.

The phase transition stage refers to the period during which ice crystals form in the freezing process. A shorter phase transition duration leads to reduced ice crystal formation time and smaller ice crystals, thereby minimizing their impact on cellular structure [5]. As shown in Figure 5b, the temperature of the samples decreases gradually during the phase transition stage due to the formation of ice crystals and the associated release of latent heat. In line with the observations in the pre-cooling stage, the AMF-40 group exhibited the shortest phase transition time of 723 ± 32.59 s, which is 31.4% shorter than that of the AMF-0 group (*p* < 0.05). The remaining groups are ranked in the following order of phase transition duration: AMF-60, AMF-80, and AMF-20. This can be attributed to the induction of an oscillating magnetic field during freezing, which enhances thermal conductivity and reduces the time spent in the phase of maximum ice crystal formation [10].

During the subcooling stage, a rapid decrease in temperature occurs, primarily due to the freezing of the majority of water within the muscle tissue. No significant difference (*p* > 0.05) was observed in the subcooling time between the groups. The shortest subcooling time was recorded in both the AMF-60 and AMF-40 groups. Overall, the use of magnetic field-assisted freezing significantly reduced both the phase transition time and the total freezing duration.

#### 3.1.2. Changes in the Microstructure of Shrimp

During the freezing process, the barrier effect of the cell membrane and variations in solute concentration cause the initial formation of crystallization nuclei outside the cells. Water within the cells migrates outward due to osmotic pressure and subsequently adheres to the external crystallization nuclei, leading to the formation of ice crystals [20]. As liquid water converts to solid ice, it expands in volume, exerting pressure on the muscle fibers. The structure of the ice crystals is preserved, and they are then subjected to freeze-drying for subsequent examination using scanning electron microscopy (SEM). Consequently, the size of the holes observed in the SEM images corresponds to the size of the ice crystals [21].

As shown in Figure 6, the AMF-0 group exhibited large, irregular pores that caused damage to the muscle fibers. The histogram in Figure 7 illustrates the distribution of pore areas. The average pore areas, ranked from smallest to largest, are AMF-40, AMF-60, AMF-80, AMF-0, and AMF-20. The difference in pore size between the AMF-0 and AMF-20 groups was minimal. Specifically, the average pore area in the AMF-40 group was 6499. 54 μm^2^, representing a 71.13% reduction compared to that of AMF-0. In the AMF-20 group, the pores remained large but were more uniform, likely due to reduced osmotic pressure associated with faster freezing. The AMF-40 group demonstrated the fewest pores and the best-preserved muscle fiber structure. Consistent with the freezing curves, the magnetic field intensity in the AMF-40 group accelerated ice crystal formation in Penaeus Japonicus and mitigated cellular structure damage. As the magnetic field intensity increased, pore size initially decreased and then increased. Sun’s experiment reached a similar conclusion: as the magnetic field intensity increases, the sample quality initially improves but subsequently deteriorates. They observed that a 60 mT magnetic field reduced the size of ice crystals and minimized damage to the cellular structure [22]. Liu et al. [23] found that magnetic fields decreased ice crystal size and mitigated muscle fiber damage.

#### 3.1.3. Moisture Characteristics at Different Magnetic Field Strengths

##### Water-Holding Capacity Analysis at Different Magnetic Field Strengths

The degree of moisture loss has been used as an indicator of structural damage caused by initial freezing, the duration of frozen storage, and the thawing process [24]. Freezing and thawing processes alter the moisture content of shrimp meat, making mass loss during thawing a significant indicator for evaluating shrimp meat quality [25].

As shown in Figure 8, the AMF-0 group exhibited the highest thawing loss, followed by the AMF-20 group. No statistically significant difference was observed between these two groups, indicating that a low Magnetic Field Strength did not significantly affect thawing loss (*p* > 0.05). The AMF-40 group exhibited the lowest thawing loss, representing a 62.1% reduction compared to the AMF-0 group. With increasing magnetic field strength, thawing loss initially decreased and then increased, suggesting that an optimal Magnetic Field Strength can minimize microstructural damage in Penaeus Japonicus. In their study, Maninder Kaur and colleagues demonstrated that the application of a magnetic field during freezing significantly influenced the thawing loss of tomatoes [3]. They observed that the magnetic field accelerated the freezing process, resulting in the formation of ice crystals and less damage to the cellular structure of the tomatoes. Similarly, studies by Chen et al. [11] and Feng et al. [26] found that magnetic field-assisted freezing reduced thawing loss in green and red chili peppers, respectively.

Cooking loss refers to the reduction of substantial amounts of liquid and small quantities of soluble material during the heating process. Partial dehydration during freezing can lead to an increase in cooking loss, as the freezing process affects the stability of the protein network and its interactions with water, thereby altering the cooking loss of the sample [27]. Figure 8 illustrates trends among the experimental groups; however, no significant differences were observed. The AMF-40 group exhibited the lowest cooking loss, showing a 34.2% reduction compared to the AMF-0 group. This reduction may be attributed to the magnetic field’s regulation of ice crystal formation and protein thermal denaturation during freezing. Gan et al. [13] reported that magnetic field-assisted freezing resulted in a shorter freezing duration, faster freezing rate, and finer ice crystals, which caused less cellular damage, and observed that magnetic field-assisted dip freezing reduced cooking loss in pork.

Centrifugal loss refers to the ability of meat to retain its initial moisture content when subjected to external pressures. During the freezing process, ice crystals form within the meat, leading to the separation of bound water and protein molecules. Furthermore, the high concentration of solutes within the cells accelerates protein denaturation, thereby decreasing the muscle’s ability to retain water [20]. Figure 8 illustrates that as Magnetic Field Strength increases, centrifugal loss initially decreases before rising again. No significant differences (*p* > 0.05) were observed among the treatments.

##### Changes in Water Distribution of Shrimp at Different Magnetic Field Strengths

Low-field nuclear magnetic resonance (LF NMR) is a widely used non-destructive technique for evaluating water distribution in food products [16]. T2b1 and T2b2 represent water molecules that are tightly bound to macromolecules, where T2b1 corresponds to firmly bound water and T2b2 to weakly bound water. T21 denotes immobile water located within the densely interconnected myofibrillar proteins. T22 refers to free water present in the interstitial spaces between fibril bundles [28].

The LF NMR curves were presented in Figure 9a. The transverse relaxation time (T2) serves as an indicator of the sample’s ability to bind water molecules: shorter T2 values reflect stronger binding affinity to water molecules. The peak area is used to assess the sample’s water content. Figure 9b–e illustrate the T2 values and peak areas corresponding to the peaks of the four types of water. As shown in Figure 9b,c, the T2 values for the strongly bound water peaks follow this order: Fresh, AMF-60, AMF-40, AMF-80, AMF-20, and AMF-0. For weakly bound water, the T2 values are ranked as follows: Fresh, AMF-40, AMF-80, AMF-60, AMF-20, and AMF-0. This variation may be attributed to the effects of denaturation, aggregation, and shrinkage of the muscle fibers [29]. The application of the magnetic field effectively mitigates the disruption of muscle fibers, thereby preserving the sample’s ability to bind water.

The increase in water mobility can be attributed to the breakdown of the tightly connected myofibrillar network during freezing. Freezing induces the disruption of some hydrogen and ionic bonds in myofibrillar proteins, leading to the exposure of hydrophobic groups as a result of mechanical damage to the protein molecules. Consequently, some of the bound water is transformed into free water, and a portion may even be lost during thawing [20]. Figure 9d demonstrates that the AMF-40 group exhibited a 10.4% higher immobilized water content compared to the AMF-0 group. The AMF-40 group showed the lowest T2 value, indicating that the magnetic field’s influence during freezing helps mitigate damage to the muscle fiber network, thereby reducing water migration. Wei et al. [30] conducted a study to examine the effect of gradient magnetic fields on freezing quality, employing low-field nuclear magnetic resonance (NMR) to analyze the water distribution within the samples. The results revealed that the application of magnetic fields significantly reduced water migration in tilapia.

Figure 9e illustrates that the AMF-40 group exhibited the lowest free water content, with a reduced conversion of immobilized water to free water. Regarding overall moisture content, the AMF-40 group closely approximated the moisture content of the fresh group, indicating that less water was lost during the freezing process. This suggests that magnetic field-assisted freezing can effectively minimize water loss. Figure 9f presents the percentages of various moisture contents. It is evident that the AMF-40, AMF-60, and AMF-80 groups contained more immobilized water and less free water, consistent with the earlier analysis of moisture distribution.

Magnetic Resonance Imaging (MRI) was utilized to visualize the moisture content in the samples, and the results were processed using pseudo-color software. The numbers represent the relative strength of the signal. It was observed that the fresh samples exhibited the highest and most uniform moisture content, with no significant decrease in moisture. In contrast, the frozen samples showed varying degrees of moisture loss, with the AMF-0 group experiencing the greatest moisture reduction, while the AMF-40 group exhibited the least moisture loss, closely approximating the moisture content of the fresh group. Hu et al. [31] and Zhou et al. [32] investigated the effect of magnetic field-assisted freezing on sample quality: MRI results from both studies indicated that the application of a magnetic field reduced water migration in the samples.

#### 3.1.4. Physicochemical Property at Different Magnetic Field Strengths

##### Color Changes Analysis at Different Magnetic Field Strengths

The freezing process damages the muscle fiber structure, causing the conversion of some immobilized water into free water, which increases the water content on the sample surface, thereby enhancing light refraction and raising the L* value. The change in L*, which defines brightness, can serve as an indicator of freezing quality.

As shown in Figure 10, the L* values increased in all experimental groups compared to the fresh control group. Among them, the AMF-0 group exhibited the highest L* value, indicating that during freezing, a portion of immobilized water was converted into free water, thereby increasing the brightness of Penaeus Japonicus. Conversely, the AMF-40 group had the lowest L* value, suggesting that a magnetic field strength of 40 G was most effective in freezing Penaeus Japonicus with minimal damage to the meat. Jiang et al. [33] reported a similar trend in their study on frozen meat thawing under varying magnetic field strengths, observing a consistent decrease in L* values as magnetic field strength increased. These findings indicate that the magnetic field effectively reduces moisture migration.

As shown in Figure 11, the ΔE value initially decreased and then increased, with the AMF-40 group exhibiting the minimum value. This indicates that an appropriate magnetic field treatment can effectively maintain the fresh color of frozen fish products. The underlying mechanism may lie in the ability of the magnetic field at this specific intensity to optimize the freezing process by promoting the formation of fine ice crystals, thereby minimizing physical damage to the tissue cells.

##### Texture Analysis at Different Magnetic Field Strengths

The texture of food products is a crucial determinant of food quality, influencing the tactile perception of consumers during purchase and serving as an important indicator of consumer acceptance [34]. This study investigates the effect of magnetic field-assisted freezing on two key textural properties: hardness and springiness. Hardness refers to a material’s ability to locally resist compression by a hard object on its surface and is mainly related to the integrity of muscle tissue and the stability of protein structures. As illustrated in Figure 12, in all magnetic field-treated groups, hardness initially increased and then stabilized, indicating that the influence of the magnetic field diminished with increasing field strength. Compared to the AMF-0 group, magnetic field-assisted freezing demonstrated a beneficial impact on freezing quality. Springiness is defined as the ability of a food product to recover its original shape or volume after compression. It is a critical indicator of texture, often correlating with taste and flavor. Changes in springiness and hardness followed similar patterns, both showing varying degrees of reduction relative to the fresh samples. Among the experimental groups, the AMF-40 group exhibited the highest springiness and hardness, which may be attributed to water loss and redistribution caused by ice crystal formation. Leng et al. [18] reported that static magnetic fields effectively inhibited decreases in hardness and chewiness during the freezing of echinoid catfish.

##### DSC Analysis

Freezing alters the structure and thermal stability of proteins [35]. The maximum denaturation temperature observed in Differential Scanning Calorimetry (DSC) corresponds to the peak temperature on the DSC curve, marking the onset of significant protein denaturation. At this temperature, the organized molecular structure begins to transition into a disordered state. The first peak, Tmax1 (45–50 °C), corresponds to the denaturation of myosin, while Tmax2 (60–65 °C) represents the denaturation of actin. Changes in the transition temperature (Tmax) and enthalpy (ΔH) serve as direct indicators of protein structural stability, with higher values indicating enhanced resistance to denaturation [21].

The denaturation temperatures and enthalpy values for myosin are shown in Figure 13a. The fresh group exhibited the highest ΔH1, indicating the greatest thermal stability among the samples. This can be attributed to the absence of freezing-induced structural damage, which preserves the integrity of protein molecules. According to data by Tan et al. [36], smaller ice crystals inflict less cellular damage, whereas larger crystals disrupt the secondary protein structure, thereby reducing thermal stability. Across all treatment groups, a trend of increasing followed by decreasing thermal stability was observed. The AMF-40 group demonstrated the highest myosin thermal stability, suggesting that the alternating magnetic field alleviated the structural damage caused by freezing.

Among all experimental groups, the ΔH2 value for the AMF-0 group was the lowest at 1.06 J/g, which was 33.8% lower than that of the AMF-40 group. A statistically significant difference (*p* < 0.05) was observed between these two groups in terms of freezing quality. As shown in Figure 13b, the ΔH2 trend mirrors that of ΔH1, indicating that the AMF-40 treatment contributes to the preservation of actin structure in frozen Penaeus Japonicus. Representative DSC thermograms for samples subjected to different freezing treatments are illustrated in Figure 13c.

### 3.2. The Effect of Magnetic Field Frequency

#### 3.2.1. Changes in Freezing Time at Different Magnetic Field Frequencies

The pre-cooling stage refers to the process of gradually cooling items from room temperature (4 °C) to near the freezing point (−1 °C). Lowering the temperature of items from room temperature to near freezing conditions creates the necessary environment for the subsequent freezing process. Within this temperature range, water molecules gradually lose their fluidity, though they have not yet fully frozen. As shown in Figure 14, the duration of the pre-cooling stage for the AMF-50 Hz group was the longest at 82 s, while both the AMF-150 Hz and AMF-200 Hz groups had a duration of 43 s, representing a 45.7% reduction compared to the AMF-50 Hz group. Magnetic fields of different frequencies influence the arrangement of water molecules. Under the influence of the magnetic field, water molecules transition from a disordered to an ordered state, aligning along the direction of the applied magnetic field. This weakens the intermolecular bonds, making it difficult for large water clusters to form, thus lowering the initial nucleation temperature and affecting the pre-cooling time. The alternating magnetic field can accelerate heat transfer, enabling rapid cooling. The periodic changes in the magnetic field cause the magnetic moments within the sample to shift accordingly, generating eddy currents. These eddy currents increase energy dissipation within the sample, leading to a rapid decrease in temperature and a subsequent shortening of the pre-cooling time.

Phase transition time refers to the duration required for a substance to change from one phase state to another during the freezing process. Specifically, it denotes the period from the formation of the initial ice nucleus to the complete growth of ice crystals. This stage is critical, as it directly affects ice crystal formation and growth, thereby influencing the quality and structural integrity of the frozen material. As shown in Table 2, the longest phase transition time was observed in the AMF-50 Hz group, lasting 556 s, whereas the shortest time of 294 s occurred in the AMF-200 Hz group. Alternating magnetic fields can regulate the freezing crystallization process of water in food matrices by promoting the formation of more orderly ice crystals, reducing the formation of large ice crystals, minimizing cellular damage, and ultimately improving the quality of frozen food. Panayampadan, A.S. et al. [37] demonstrated that an alternating magnetic field of 7.02 mT reduced phase transition time during the freezing of guava blocks and minimized juice loss after thawing, thereby better preserving the original texture characteristics. Furthermore, alternating magnetic fields enhance heat transfer efficiency, enabling samples to reach phase transition temperatures more rapidly and thus shortening phase transition time. Additionally, alternating magnetic fields can modify the cellular structure within samples by reducing intercellular gaps and promoting a more uniform fiber distribution, further influencing phase transition time.

#### 3.2.2. Physicochemical Property at Different Magnetic Field Frequencies

##### Color Changes Analysis at Different Magnetic Field Frequencies

The flesh of Penaeus Japonicus is typically white and tender. The L* value, measured using a color difference meter, represents brightness, with higher values indicating greater brightness. In this study, the appearance of the frozen samples remained largely unchanged, and variations in L* values may be related to the free water content in the shrimp samples. Higher moisture content generally results in a more moist surface, enhancing light reflectance and thereby increasing brightness. As shown in Figure 15, the L* value gradually decreases with increasing magnetic field frequency, leading to a reduction in brightness. The extent of this decrease varies across different frequency ranges, with the largest reduction (~4.7%) occurring between the AMF-150 Hz and AMF-200 Hz groups. Overall, the L* value decreases from 55.84 at AMF-50 Hz to 49.79 at AMF-250 Hz, representing an approximate 10.8% decline. Significant differences in brightness were observed between the AMF-200 Hz and AMF-250 Hz groups compared to the AMF-50 Hz and AMF-100 Hz groups. This phenomenon may be attributed to the influence of the magnetic field on water molecule movement, which alters the secondary structure of proteins, subsequently affecting water binding and migration. During spoilage of Penaeus Japonicus, the flesh color gradually darkens and may eventually turn black. This color change is primarily caused by protein degradation and microbial activity. Protein degradation generates metabolic byproducts that alter the chemical composition and structure of the shrimp meat, leading to color changes. Furthermore, microbial metabolism produces pigments that contribute to further darkening of the flesh.

As depicted in Figure 16, the change in ΔE initially decreases and then increases, with the minimum value observed in the AMF-200 Hz group. YE et al. [38] examined the impact of different alternating magnetic field frequencies (50 Hz, 100 Hz, 150 Hz, 200 Hz, 250 Hz) on the freezing quality of tilapia. Their study concluded that varying magnetic field frequencies exert distinct effects on the freezing quality of tilapia, with 200 Hz frequency leading to the most notable improvement in the quality of frozen products.

##### Texture Analysis at Different Magnetic Field Frequencies

As the freshness of Penaeus Japonicus decreases, its hardness gradually diminishes, resulting in a softer texture. As illustrated in Figure 17, no significant differences were observed between the groups. In general, hardness tends to increase with increasing frequency, although a slight decrease is observed at 250 Hz. The AMF-200 Hz group exhibited an 8.3% increase in hardness compared to the AMF-50 Hz group. This suggests the presence of an optimal magnetic field frequency that could substantially enhance the freezing quality of Penaeus Japonicus. Guo et al. [39] examined the effects of low-frequency magnetic fields and high pH values on the quality of pork myofibrillar protein gels. Their results indicated that the combined treatment of low-frequency magnetic fields and high pH values significantly improved the hardness and elasticity of pork myofibrillar protein gels, highlighting the role of magnetic field treatment in preserving the texture of aquatic products.

Springiness refers to the ability of Penaeus Japonicus meat to recover its original shape after the application of external force. It is a crucial indicator in evaluating shrimp texture, reflecting both elasticity and toughness. Shrimp exhibiting high Springiness retain superior texture and structural integrity during cooking and consumption. This property is influenced by multiple factors, including the formation and size of ice crystals during freezing, freezing rate, protein–water interactions, and Magnetic Field Strength. As shown in Figure 17, Springiness follows a trend similar to that of Hardness, with the highest value observed in the AMF-200 Hz group. This suggests that a frequency of 200 Hz provides the most favorable effect on the texture quality of frozen–thawed Penaeus Japonicus among all experimental conditions.

##### pH Analysis

The pH value serves as an indicator of the freshness of aquatic products. Rapid freezing helps minimize cellular damage caused by ice crystal formation, thereby influencing pH changes. The survival pH range for Penaeus Japonicus is between 7.0 and 8.5. As shown in Figure 18, while no significant differences were observed between the groups, there was a noticeable trend in pH changes. Specifically, as the frequency increased, the extent of pH reduction became smaller. This may be attributed to protein denaturation during freezing, which releases hydrogen ions, or to the breakdown of glycogen, loss of water and soluble substances, and the production of acidic byproducts during thawing. These processes can all contribute to pH decrease. However, rapid freezing produces smaller ice crystals, which cause less damage to cellular structure and mitigate the associated pH reduction.

#### 3.2.3. Moisture Characteristics at Different Magnetic Field Frequencies

##### Water-Holding Capacity Analysis at Different Magnetic Field Frequencies

As shown in Figure 19, an increase in the frequency of the alternating magnetic field led to a gradual decrease in thawing loss across all experimental groups. The AMF-250 Hz group exhibited the lowest thawing loss at 0.52%, whereas the AMF-50 Hz group had the highest thawing loss at 1.03%. The AMF-250 Hz group demonstrated a 49.5% reduction in thawing loss compared to the AMF-50 Hz group. According to the analysis of significant differences, under low-intensity magnetic field conditions, the frequency of the alternating magnetic field did not significantly affect thawing loss. Otero, L et al. [40] investigated the impact of different frequencies (20 Hz, 50 Hz, 200 Hz, 2000 Hz) of a weak oscillating magnetic field (OMF) at 0.8 mT on the freezing process of pure water and 0.9% NaCl solution. The study concluded that varying oscillation frequencies did not significantly influence supercooling, nucleation time, or freezing kinetics. Although magnetic field strength did affect the freezing process, the frequency’s impact was not significant, likely due to the magnetic field strength being outside the optimal range.

High-freshness Penaeus Japonicus lose less moisture during steaming, resulting in a relatively stable protein structure and minimal steaming loss. In contrast, low-freshness shrimp, which have already lost some moisture during storage, experience protein denaturation, leading to greater steaming loss. As shown in Figure 19, no clear trend is observed in the cooking losses across the experimental groups. The highest cooking loss occurred in the AMF-150 Hz group at 16.20%, while the lowest was observed in the AMF-200 Hz group at 11.22%. A significant difference was found between the AMF-200 Hz group and the other experimental groups, while no significant differences were observed among the remaining groups. YE et al. [38] also demonstrated that magnetic fields could enhance the quality of frozen tilapia, with 200 Hz showing the most notable improvement in frozen product quality.

In terms of centrifugal loss, the AMF-100 Hz group exhibited the highest centrifugal loss at 13.84%, closely followed by the AMF-50 Hz group at 13.72%, with a minimal difference between the two. The centrifugal losses for the AMF-150 Hz, AMF-200 Hz, and AMF-250 Hz groups were 11.65%, 11.25%, and 11.20%, respectively. This could be attributed to the fact that at lower alternating magnetic field frequencies, the rate of change in the magnetic field is slower, leading to relatively minor effects on the water content and cellular structure of Penaeus Japonicus. In this frequency range, the magnetic field likely influences the permeability of the cell membrane, promoting the exchange of substances between the interior and exterior of the cell, thereby enhancing the water retention capacity of the cells and reducing centrifugal loss. However, as the alternating magnetic field frequency increases beyond a certain point, the rate of field changes accelerates, potentially resonating with the vibration frequency of water molecules within the shrimp. This resonance effect allows water molecules to bind more effectively with the cells, further improving cellular water retention and minimizing centrifugal loss. Despite these observations, no significant differences were found within the experimental frequency range.

##### Changes in Water Distribution of Shrimp at Different Magnetic Field Frequencies

The typical NMR spectra for each group are shown in Figure 20a. The application of a 200 Hz alternating magnetic field significantly enhanced the water distribution in Penaeus Japonicus, with the proportion of bound water notably higher than that in the AMF-50 Hz group. Figure 20b displays the proportions of the three types of water across the five groups of samples. The combined content of strongly bound water and weakly bound water in the AMF-150 Hz and AMF-200 Hz groups was 5%, a significant improvement compared to the AMF-50 Hz group. This suggests that the 200 Hz alternating magnetic field can effectively improve the freezing quality. Among all experimental groups, the AMF-200 Hz group exhibited a 2% higher bound water content than the AMF-50 Hz group, indicating that an optimal magnetic field frequency can reduce water mobility. No significant differences were observed in the free water content among the experimental groups.

After processing the images with pseudo-color software, darker colors were observed to correspond to higher moisture content at the respective sample locations. Analysis of these images reveals that the moisture distribution in the AMF-50 Hz and AMF-250 Hz groups differs slightly from that in the other groups. Specifically, the AMF-150 Hz and AMF-50 Hz groups exhibited a uniform moisture distribution, with no significant moisture loss detected around the edges. However, post-freezing, all samples showed varying degrees of edge moisture loss, with the SMF-0 group demonstrating the most pronounced reduction, distinctly differing from the other groups. Hu et al. [31] investigated the effects of electromagnetic field-assisted freezing on pork quality, reporting that the application of a magnetic field significantly enhanced sample quality.

## 4. Conclusions

The findings reveal that the magnetic field expedites the freezing process, with AMF-40 exhibiting the most significant impact among the different magnetic field strengths tested. In comparison to other freezing methods, AMF-40 reduces water loss during thawing and cooking, while maintaining the color, water-retention capacity, and textural attributes of the frozen shrimp meat. Additionally, AMF-40 effectively hinders the migration and loss of both bound and free water. Observations using an electron microscope confirm that AMF-40 generates the smallest and most uniformly sized ice crystals. However, there were no statistically significant differences observed among AMF-40, AMF-60, and AMF-80, indicating the necessity for expanding the range of magnetic field intensities to detect more distinct trends in freezing Penaeus Japonicus. In terms of magnetic field frequency experiments, AMF-200 Hz notably reduces the overall freezing duration and decreases water loss during thawing and cooking while upholding the hardness and Springiness of Penaeus Japonicus, thereby enhancing its texture. Furthermore, the AMF-200 Hz magnetic field improves the water-holding capacity of Penaeus Japonicus, diminishes water loss, and restrains the decline in bound water content. These results suggest that magnetic fields offer a novel freezing technology with the potential to enhance the quality of frozen Penaeus japonicus. It should be noted that the present study did not examine the effects of Magnetic Field Strength on pH, which may constrain a complete comparison with frequency-related effects. Furthermore, the advancement of magnetic field-assisted freezing devices is crucial for the commercial application of this technology and for improving the quality of frozen products.

## Figures and Tables

**Figure 1 foods-14-04112-f001:**
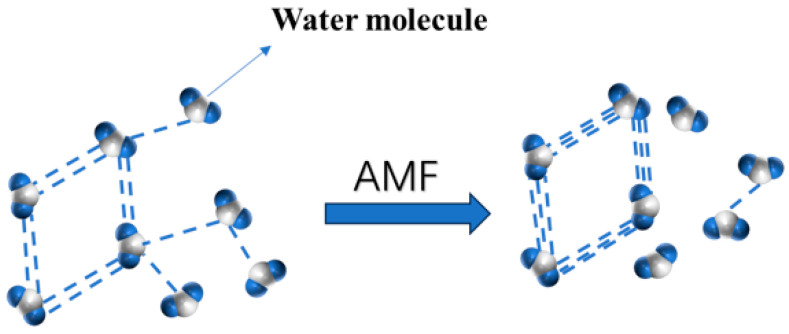
Certain hydrogen bonds are reinforced, while others are disrupted. (dash-line indicate hydrogen bonds.)

**Figure 2 foods-14-04112-f002:**
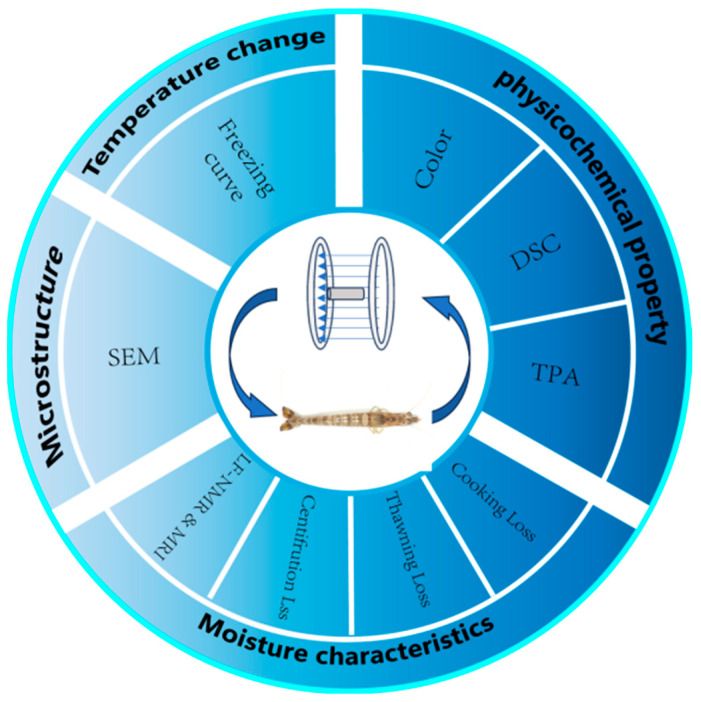
The experimental design.

**Figure 3 foods-14-04112-f003:**
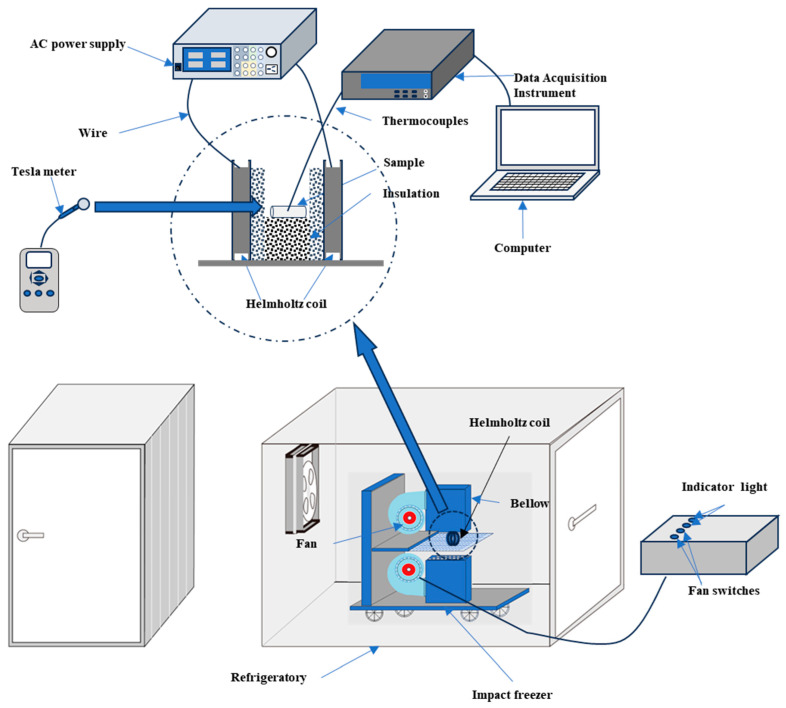
Schematic of the magnetic field freezer.

**Figure 4 foods-14-04112-f004:**
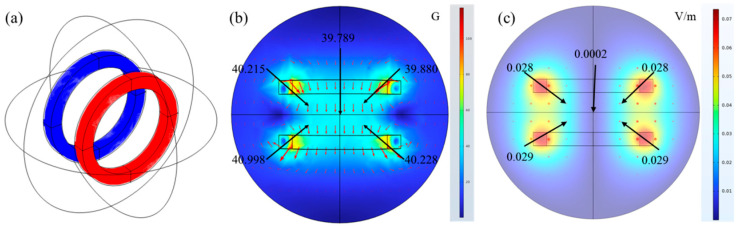
The model of the Helmholtz coil and the magnetic field strength. (**a**) Schematic representation of the Helmholtz coil; (**b**) schematic representation of the strength of the Magnetic field at the shrimp freeze position; (**c**) schematic representation of the strength of the electric field at the shrimp freeze position.

**Figure 5 foods-14-04112-f005:**
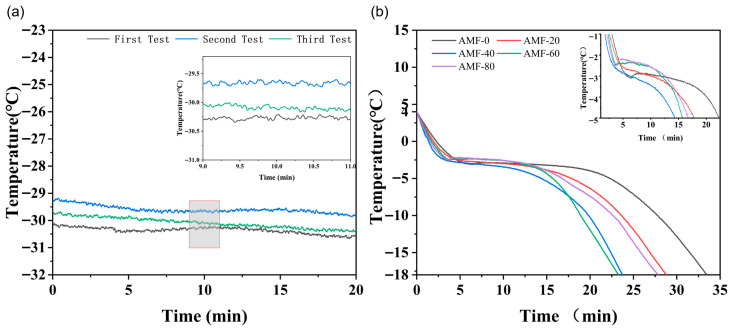
Temperature changes before and after switching the magnetic field (**a**) and freezing curve (**b**).

**Figure 6 foods-14-04112-f006:**
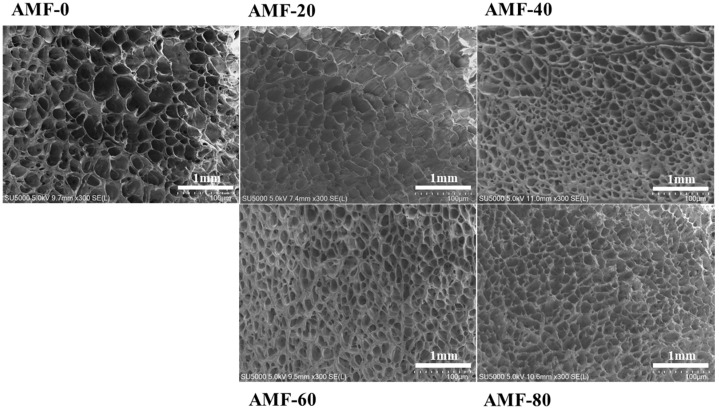
SEM images of shrimp frozen by different treatments.

**Figure 7 foods-14-04112-f007:**
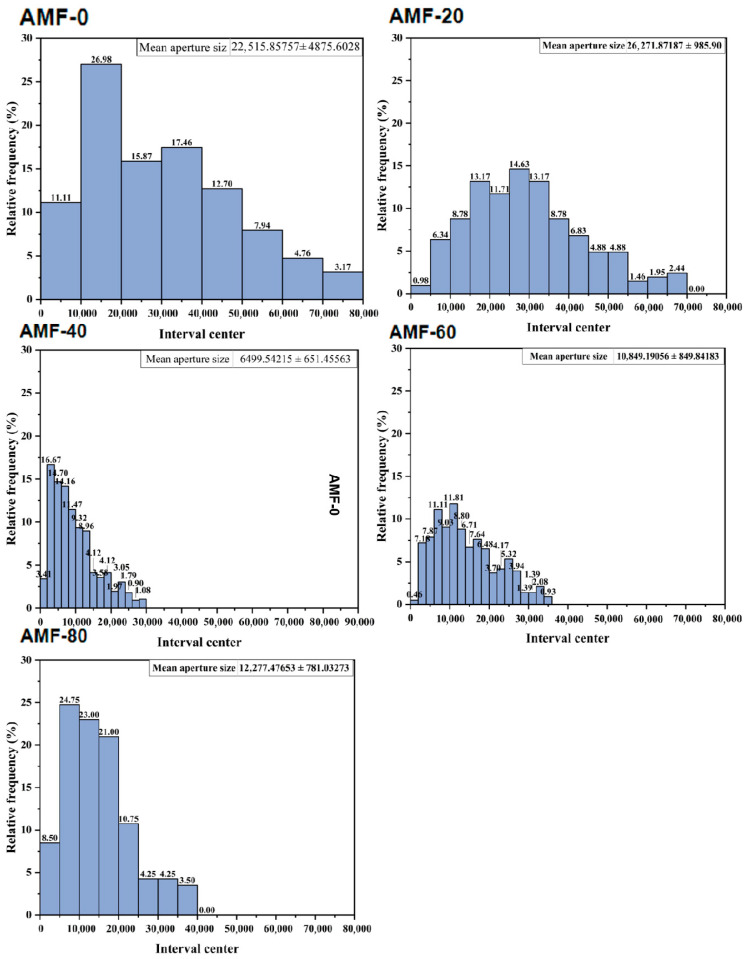
The distribution of the pore area.

**Figure 8 foods-14-04112-f008:**
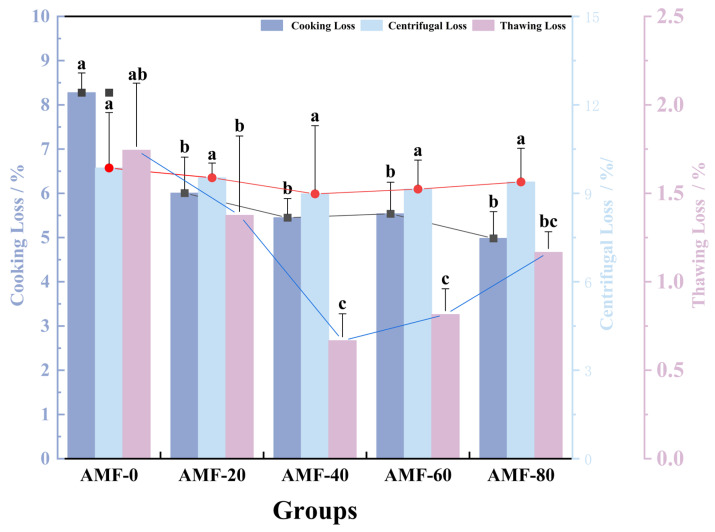
Analysis of Cooking loss (%), centrifugation loss (%) and thawing loss (%) at Different Magnetic Field Strengths. Note: Repeat the experiment 3 times for each group and different lowercase letters indicate significant differences (*p* < 0.05).

**Figure 9 foods-14-04112-f009:**
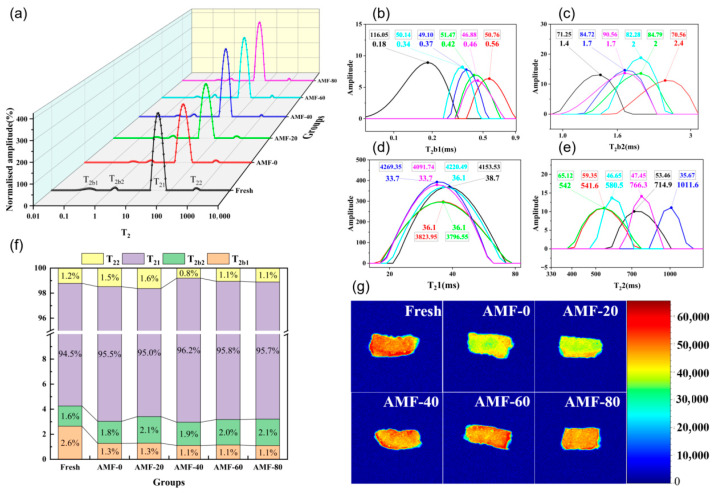
(**a**) LF NMR curve; (**b**) localized magnification of the first peak on LF NMR; (**c**) localized magnification of the second peak on LF NMR; (**d**) localized magnification of the third peak on LF NMR; (**e**) localized magnification of the fourth peak on LF NMR; (**f**) relative content of water components (%); (**g**) MRI image of shrimp frozen by different treatments.

**Figure 10 foods-14-04112-f010:**
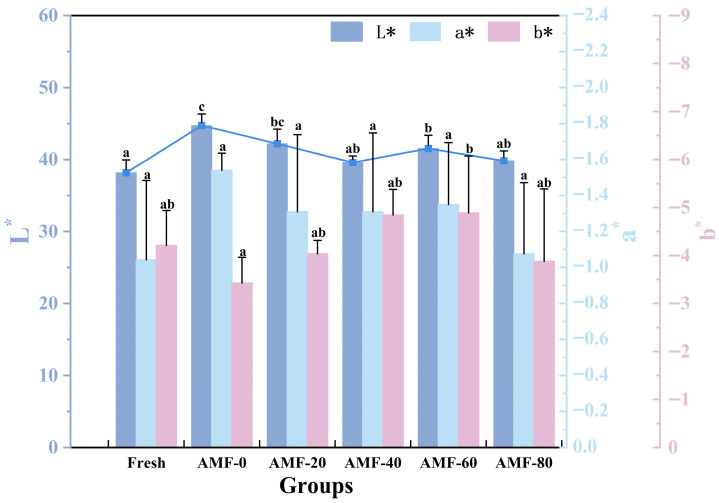
Color analysis. Note: Repeat times for each group and different lowercase letters indicate significant differences (*p* < 0.05).

**Figure 11 foods-14-04112-f011:**
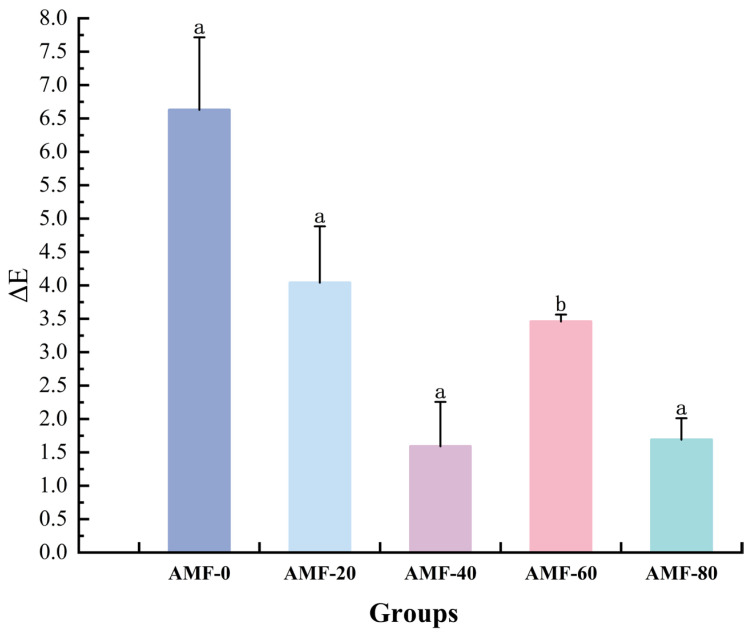
ΔE analysis of samples treated with different magnetic field strength. Note: different lowercase letters indicate significant differences (*p* < 0.05).

**Figure 12 foods-14-04112-f012:**
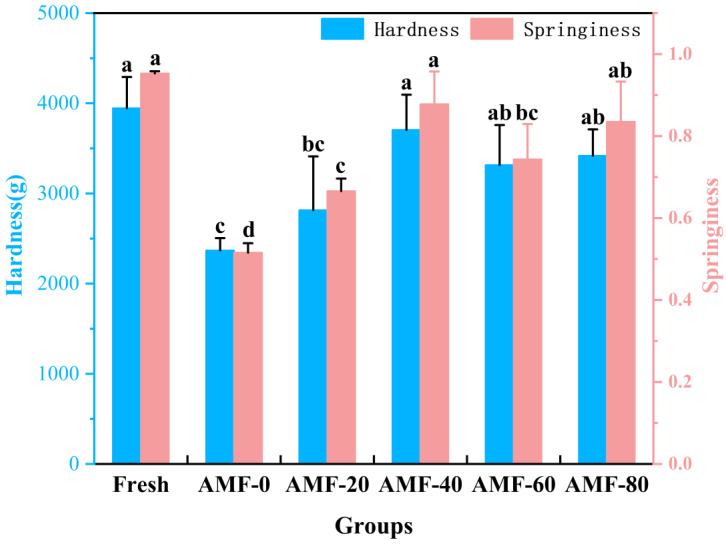
Texture analysis results. Note: Repeat 3 times for each group and different lowercase letters indicate significant differences (*p* < 0.05).

**Figure 13 foods-14-04112-f013:**
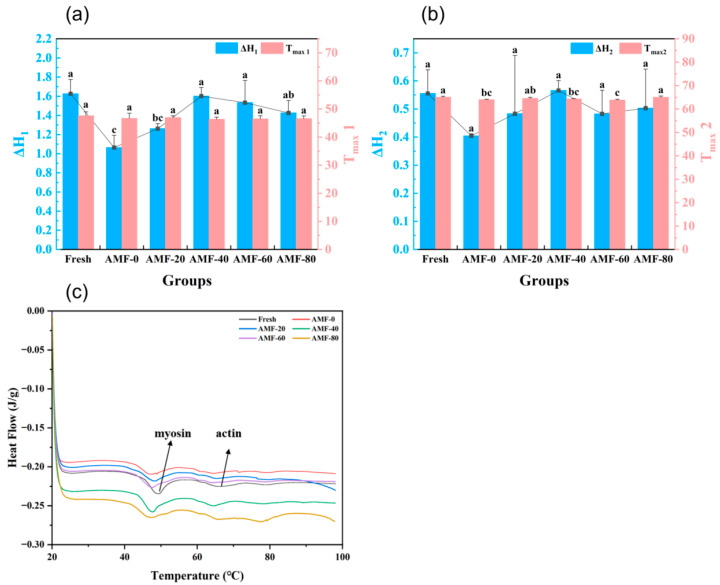
(**a**) DSC analysis of peak 1; (**b**) DSC analysis of peak 2; (**c**) Differential scanning calorimetry (DSC) scanning curves of the Penaeus Japonicus muscle after different freezing treatments. Note: Repeat 3 times for each group and different lowercase letters indicate significant differences (*p* < 0.05).

**Figure 14 foods-14-04112-f014:**
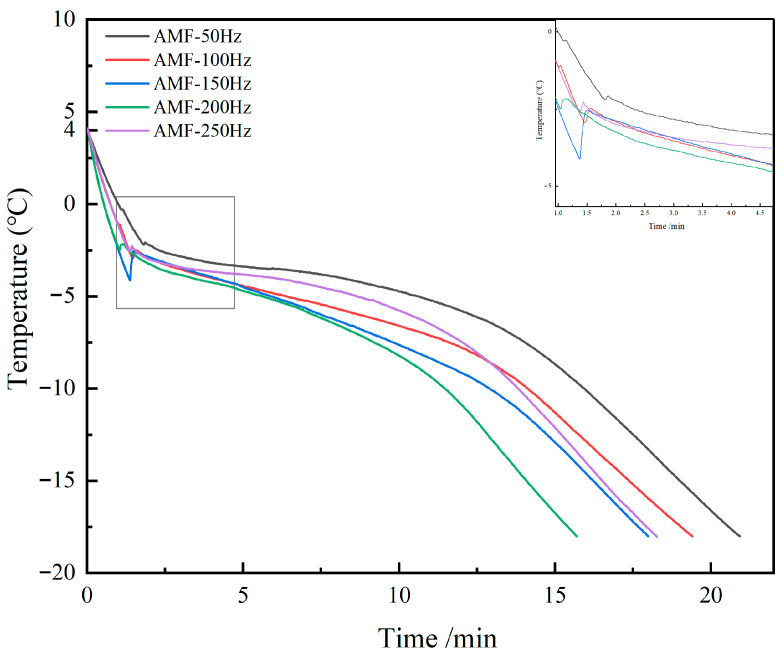
Temperature change curve of Penaeus Japonicus during freezing.

**Figure 15 foods-14-04112-f015:**
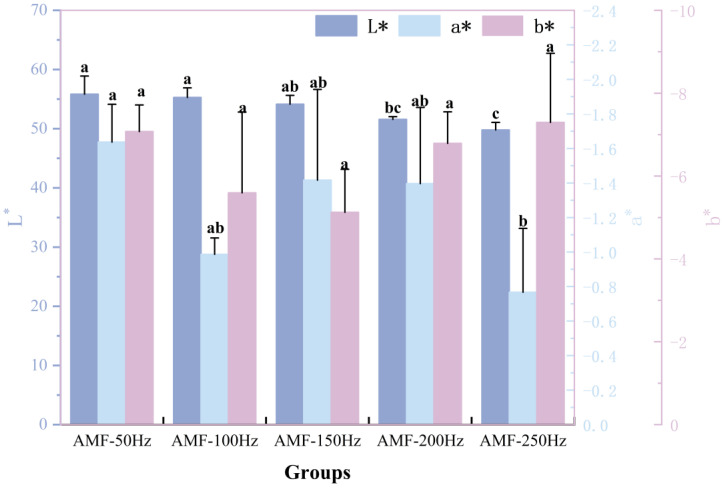
Color difference analysis of samples treated with different magnetic field frequencies. Note: different lowercase letters indicate significant differences (*p* < 0.05).

**Figure 16 foods-14-04112-f016:**
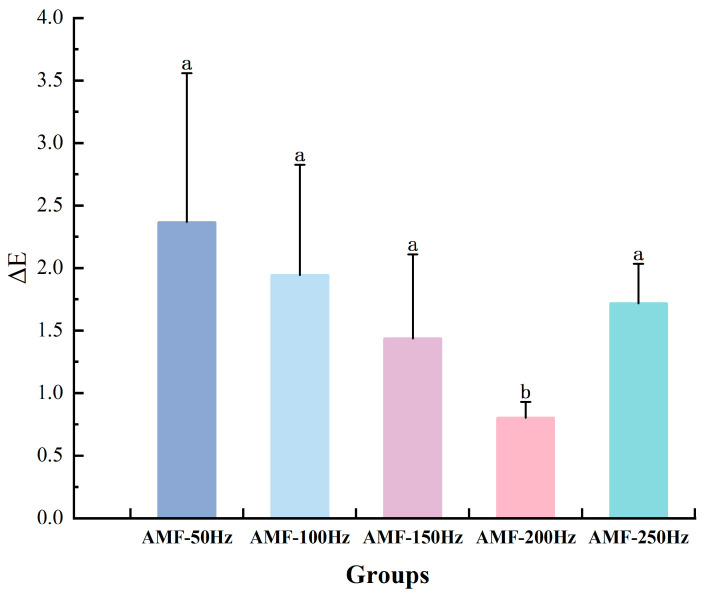
ΔE analysis of samples treated with different magnetic field frequencies. Note: different lowercase letters indicate significant differences (*p* < 0.05).

**Figure 17 foods-14-04112-f017:**
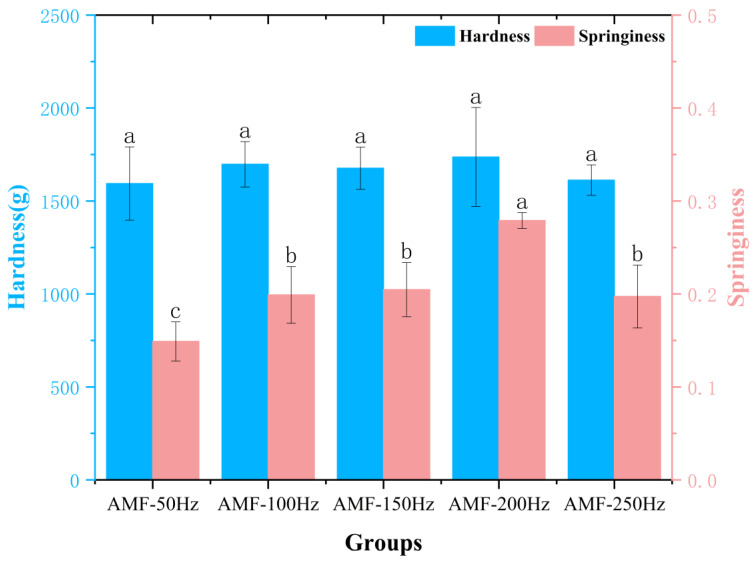
Results of Hardness and springiness of samples after treatment with different magnetic field frequencies. Note: different lowercase letters indicate significant differences (*p* < 0.05).

**Figure 18 foods-14-04112-f018:**
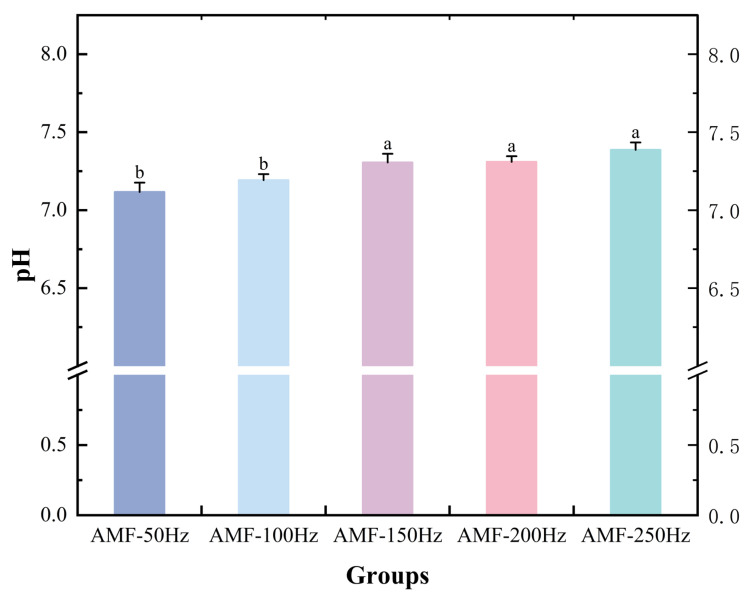
Results of pH of samples after treatment with different magnetic. Note: different lowercase letters indicate significant differences (*p* < 0.05).

**Figure 19 foods-14-04112-f019:**
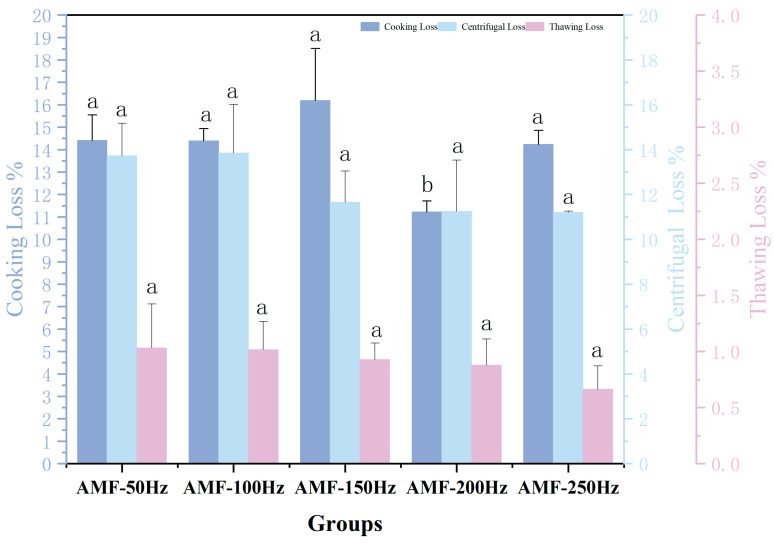
Analysis of Cooking loss (%), centrifugation loss (%) and thawing loss (%) at Different Magnetic Field Frequencies. Note: Repeat the experiment 3 times for each group and different lowercase letters indicate significant differences (*p* < 0.05).

**Figure 20 foods-14-04112-f020:**
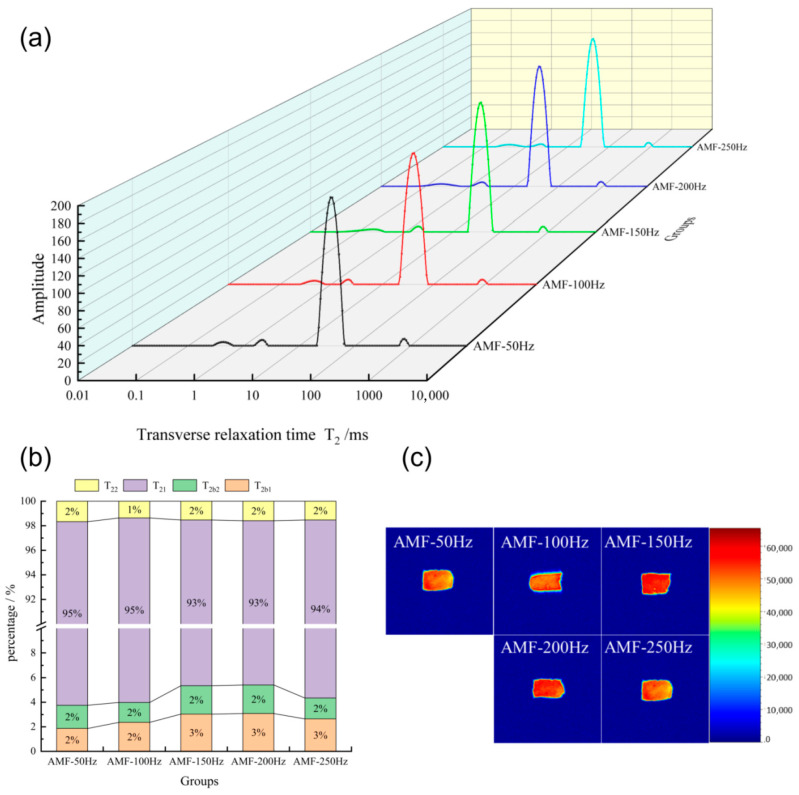
(**a**) LF NMR curve; (**b**) relative content of water components (%); (**c**) MRI image of shrimp frozen by different treatments.

**Table 1 foods-14-04112-t001:** Time spent in each phase for each treatment group.

Treatment	Pre-Cooling Time/s	Phase Transition Time/s	Subcooling Time/s	Total Freezing Time/s
AMF-0	179 ± 4.72 ^a^	1054 ± 88.26 ^a^	549 ± 108.89 ^a^	1782 ± 197.18 ^a^
AMF-20	141 ± 14.19 ^b^	891 ± 51.16 ^b^	539 ± 105.69 ^a^	1571 ± 152.21 ^a^
AMF-40	112 ± 18.23 ^c^	723 ± 32.59 ^c^	646 ± 69.97 ^a^	1482 ± 59.66 ^a^
AMF-60	135 ± 10.69 ^b^	832 ± 53.41 ^b^	500 ± 42.93 ^a^	1469 ± 92.08 ^a^
AMF-80	130 ± 24.34 ^b^	844 ± 51.97 ^b^	553 ± 131.19 ^a^	1528 ± 198.16 ^a^

Note: Repeat the experiment 3 times for each group and different letters (a–c) represent significant difference (*p* < 0.05) between different samples in the same column.

**Table 2 foods-14-04112-t002:** Time spent in each phase for each treatment group.

Groups	Pre-Cooling Time/s	Phase Transition Time/s	Subcooling Time/s	Total Freezing Time/s
AMF-50 Hz	82	556	618	1256
AMF-100 Hz	59	335	780	1164
AMF-150 Hz	43	311	725	1079
AMF-200 Hz	43	294	605	942
AMF-250 Hz	58	298	740	1096

## Data Availability

The original contributions presented in the study are included in the article. Further inquiries can be directed to the corresponding author.

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
