# Peer review of "Foods2025, 14(23), 4112;https://doi.org/10.3390/foods14234112"

_foods, 2025, doi:10.3390/foods14234112_

Round 1

Reviewer 1 Report

Comments and Suggestions for Authors

Ice crystals size and distribution influence the quality of freezed foods. Because of its physical effects alternating magnetic field applied during phase change can assist to achieve smaller and more uniform ice crystals, less tissue damage, or improved water-holding capacity, for instances. Different magnetic field intensities (0–80 G) and frequencies (50–250 Hz) were tested using a specially designed impact freezer. Therefore the topic of the manuscript can be considered as interesting and can have relevance for the practice as well.

The manuscript is well structured. Introduction summarized well the background and relevance of the study. The study employs diverse analytical techniques (SEM, LF-NMR, DSC, texture analysis, and colorimetry). The novelty of the study is clearly given. Microstructural SEM analysis revealed smaller and more uniform ice crystals under optimal AMF conditions. LF-NMR and MRI confirmed enhanced water-holding capacity. Results showed that a magnetic intensity of 40 G (AMF-40) and frequency of 200 Hz (AMF-200Hz) produced the best results/quality in terms of reduced freezing time, thawing and cooking loss.  The manuscript contains valuable results that are discussed with relevant references, but need revision before publishing (see my comments below).

Comments, suggestions:

The generalizability of the conclusions drawn from the results is questionable, since—if I understand correctly—only one batch was used for the tests (should be ’biological’ replicates as well).

Please give how was the magnetic field intensity range selected/determined.

Why did the authors use just 3 days freeze storage after magnetic field pretreatments (oxidation, enzymatic change occur during longer storage)?

Please improve the visibility of Figure 5 and Figure 7 (mainly labels,axis titles and units).

Please clearly indicate the fixed parameter values for the figures/tables, as well.

Please check the typing errors (see in line 607 ’ Figureure’; use of ’Japonicus’-capital, for examples).

Reviewer 2 Report

Comments and Suggestions for Authors

After reviewing the manuscript entitled “Effect of alternating magnetic field-assisted freezing process on the quality of frozen Penaeus japonicus”, I would like to provide the following comments and suggestions for improvement:

  1. Line 41: Wang et al. is missing from the reference list. Please add the missing citation and replace it with the corresponding reference number. Adjust the numbering of subsequent citations accordingly.
  2. Citation style: The current in-text citation format, which omits author names, results in awkward phrasing. For example:
    • line 319 – “[23] found that magnetic fields…”
    • line 433 – “[33] reported a similar trend…”
    • line 456 – “[18] reported that static magnetic fields…”
      Many similar cases occur throughout the manuscript. Moreover, citation usage is inconsistent, e.g.:
    • line 341 – “Similarly, studies by Chen et al. and Feng et al. found that magnetic field-assisted freezing reduced thawing loss in green and red chili peppers, respectively [10,26].”
      Please ensure a consistent and grammatically correct citation format throughout the text.
  3. It is unclear why the analyses of ΔE and pH were not conducted for the effect of magnetic field intensity, while they were performed for magnetic field frequency. Please clarify or justify this methodological decision.
  4. Line 149: The superscript 3 appears unnecessary; it would be clearer to indicate mm after each numeric value instead.
  5. Figure 16: The caption does not accurately correspond to the figure content. Please revise for consistency.
  6. Figure 18: The necessity of including a legend should be reconsidered, as legends are not provided for other figures with multiple Y-axes.
  7. Conclusions: Please verify whether the conclusions indeed refer to Japanese eel (line 706), as this appears inconsistent with the study subject (Penaeus japonicus).

Additional comments:

  • Please specify the number of shrimp used for each combination of magnetic field intensity and frequency. This information is not clearly defined in lines 223–224.
  • Section 2.4 should be supplemented with detailed freeze-drying parameters and the specifications of the lyophilizer used.

Round 2

Reviewer 2 Report

Comments and Suggestions for Authors

The authors complied to all the comments contained in the first review.

I am satisfied with the form of changes and responses.

In my opinion, the article may be published in Foods

Author Response

Thank you very much for your valuable feedback. It has been very helpful to me.